# Derivation and simulation of a computational model of active cell populations: How overlap avoidance, deformability, cell-cell junctions and cytoskeletal forces affect alignment

Vivienne Leech[1], Fiona N. Kenny[2], Stefania Marcotti [2], Tanya J. Shaw[2], Brian M. Stramer[2], Angelika Manhart [1,3]*

**1** Department of Mathematics, University College London, London, United Kingdom, **2** Randall Centre for Cell and Molecular Biophysics, King's College London, London, United Kingdom, **3** Faculty of Mathematics, University of Vienna, Vienna, Austria

* angelika.manhart@univie.ac.at

## Abstract

Collective alignment of cell populations is a commonly observed phenomena in biology. An important example are aligning fibroblasts in healthy or scar tissue. In this work we derive and simulate a mechanistic agent-based model of the collective behaviour of actively moving and interacting cells, with a focus on understanding collective alignment. The derivation strategy is based on energy minimisation. The model ingredients are motivated by data on the behaviour of different populations of aligning fibroblasts and include: Self-propulsion, overlap avoidance, deformability, cell-cell junctions and cytoskeletal forces. We find that there is an optimal ratio of self-propulsion speed and overlap avoidance that maximises collective alignment. Further we find that deformability aids alignment, and that cell-cell junctions by themselves hinder alignment. However, if cytoskeletal forces are transmitted via cell-cell junctions we observe strong collective alignment over large spatial scales.

## Author summary

Collective dynamics is the study of how the interaction of individual animals, humans, cells, etc. can lead to patterns on a scale much larger than the individuals themselves. Prominent examples are flocking birds, schools of fish or the patterns of bacterial colonies. Since the behaviour of the group is very difficult to intuit from individual interaction rules, mathematical models are invaluable in testing hypotheses, making predictions, and suggesting explanations. In our collaborative work between mathematicians and biologists, we use a mathematical model, whose ingredients are motivated by the experimental observation of fibroblasts. Fibroblasts are cells that are part of the connective tissue in animals and play an important role e.g. in wound healing. Under certain conditions, e.g. in scar tissue or near tumours, these spindle-shaped cells form strongly aligned populations on large spatial scales for unclear reasons. By simulating our mathematical model, we predict how the cells' self-propulsion, overlap avoidance, deformability, and other

**Data Availability Statement:** All code written in support of this publication is publicly available at

https://github.com/angelikamanhart/Code_Alignment_Ellipses.

**Funding:** This work was supported by the Engineering and Physical Sciences Research Council (grant numbers EP/N509577/1 to VL and EP/T517793/1 to VL) which paid the salary of VL, the Wellcome Trust (grant number 107859/Z/15/Z to BS) which paid the salary of FK and SM, the European Research Council (ERC) under the European Union's Horizon 2020 research and innovation program (grant agreement no. 681808 to BS) and BBSRC project grant (BB/V006169/1, to BS), which paid the salary of SM. The funders had no role in study design, data collection and analysis, decision to publish, or preparation of the manuscript.

**Competing interests:** The authors have declared that no competing interests exist.

interactions influence the alignment dynamics. The results help to understand alignment of cell populations and apply to many other cells or organisms.

# 1 Introduction

## The challenge of active particles

The ability of particles to align with their neighbours is observed in many contexts and in many scales in biology. Famous examples include flocking of birds, schools of fish or even motion of large groups of people [1–3]. Here we focus on alignment on a cellular scale. Cell alignment has been observed in bacterial swarms [4], e.g. in myxobacteria [5], as well as in amoeboid cells [6], or fibroblasts [7]. Alignment of different types of particles has also been observed and modelled in physics [8–10], however alignment in cells is more challenging to understand, since cells can exhibit much more complicated behaviour than passive matter, see e.g. reviews [11, 12]. In particular, cells can actively self-propel, can change their shape, can interact biochemically via signalling and mechanically via physical junctions with their neighbours and with the extracellular matrix, with several levels of potential feedback involved. Physically, these systems can be thought of as consisting of agents which are able to convert energy into movement. The ability to do this, combined with interactions between neighbouring particles can give rise to collective behaviour, often in the form of alignment. The novelty of this is that energy input into the system is on a local scale, which pushes the system out of equilibrium leading to changes which can propagate through the whole system and lead to emergent structures and behaviour. Understanding the details of how this works, and looking for theoretical descriptions of living matter has been of interest for a number of years [13].

## Causes of alignment

Larger, more complex organisms, such as birds or fish, are typically able to perceive their neighbours using senses such as sight and can adjust their own direction and speed correspondingly. Some interesting work for causes of alignment for such species include [14, 15]. Cells, the main focus of this work, can perceive their surroundings in several different ways. The main ones include chemotaxis, durotaxis, signalling or mechanics. Several experimental works [16, 17] have shown that e.g. the extracellular matrix (ECM) can transmit forces, aiding cytoskeletal alignment. This has been modelled e.g. in [18, 19]. Further, the shape of the environment itself can influence alignment by introducing constraints to cell orientation on the boundary [7]. Finally, contact-based alignment can be caused by particles avoiding overlap. For movement in e.g. a fluid this would be a hard constraint, while for crawling cells "overlap" could imply cells moving on top of each other.

## Basic types of alignment models

Dynamics and patterns emerging from the interactions of many individuals are difficult to intuit from microscopic interaction rules, hence mathematical modelling and simulations are a powerful tool to shed light on the involved mechanisms. One common model type is continuum models, where the system is described in terms of continuous space and time dependent macroscopic quantities like cell density, mean direction, etc. The book [20] offers an excellent overview of models and biomedical applications. Continuum models have the advantage of ease of analysis and have been used extensively to investigate alignment and pattern formation in cell populations, for some examples see [21–27]. Agent-based models, on the other hand,

where cells are discrete objects ("agents") and each cell is equipped with its own set of equations, are particularly suitable for mechanistic hypothesis testing, since biological assumptions can be translated in a relatively straight-forward way (see e.g. reviews [28, 29]). Amongst the classical agent-based models for flocking ([30, 31]), the most famous agent-based alignment model is probably the Vicsek model [32]. This model assumes self-propelling particles align their orientation with their neighbours and produces large scale alignment whenever the alignment force is large compared to the orientational noise. While the Vicsek model and its variants have been applied to many biological problems [33, 34], it isn't suitable to test mechanisms of alignment, since alignment is already a model ingredient.

### Shape deformations

For individual cells or small groups of cells, there exist several modelling frameworks capable of describing cell shapes in a flexible and biologically well-motivated manner. Prominent approaches include 1. Phase-field models [35–39], where the cell in- and outsides are characterised by a continuous, but steep phase-field variable, 2. Models using the immersed boundary method [40–42], where the cell boundary is an explicit curve or surface interacting with the surrounding fluid, 3. Cellular Potts models [43–46], where each cell is a collection of pixels, whose dynamics follow an energy minimization, or 4. Vertex models [47–50], which describe sheets of cells via cell-cell boundaries. While phase-field and immersed boundary models allow for the description of a large class of cell shapes, they are also computationally costly and hence less suitable to investigate large numbers of cells. Cellular Potts models are not as computationally costly as they used to be (see e.g. Morpheus [51]), however the fact that they are lattice based makes analytical studies as well as the inclusion of mechanical effects more challenging. The effect of self-propulsion and cell-cell adhesions in Cellular Potts models have been investigated e.g. in [52], however it is unclear how the actin cables in Sec 4 could be included in this framework. Vertex models, on the other hand, are mostly used for simulating tissue dynamics and are less suitable for describing individual cell movement. An example that explores cell alignment in vertex models includes the work in [50]. This is a suitable approach in the context of epithelial monolayers, however the model varies from ours in the fact that cell overlap and cell-cell collisions are not included. A similar approach to ours is described in [53, 54], where populations of deformable ellipsoidal cells in 3D are modelled, which experience changes in aspect ratio as a result of cell overlap.

### This work and paper overview

In this work, we will mathematically model alignment of collectives of cells moving in two space dimensions (2D). As the main cause of alignment, we assume cells want to avoid overlap in a tuneable (i.e. non-perfect) way (Sec 2). Motivated by biologically observed, distinct behaviour of different populations of fibroblasts [55], we will assess the influence of the following characteristics of active matter on alignment: Self-propulsion (Sec 2), deformability (Sec 3), cell-cell junctions (Sec 4) and cytoskeletal forces (Sec 4).

## 2 The base model: Self-propulsion and overlap avoidance

### 2.1 Biological background & model ingredients

**Experimental motivation.** While alignment processes in active particles are relevant in many contexts, we will focus on the particular example of fibroblasts. Fibroblasts are cells in the connective tissue in animals and are responsible for making and remodelling the ECM. Fibroblast and ECM alignment is observed during various scarring pathologies. In [55] we

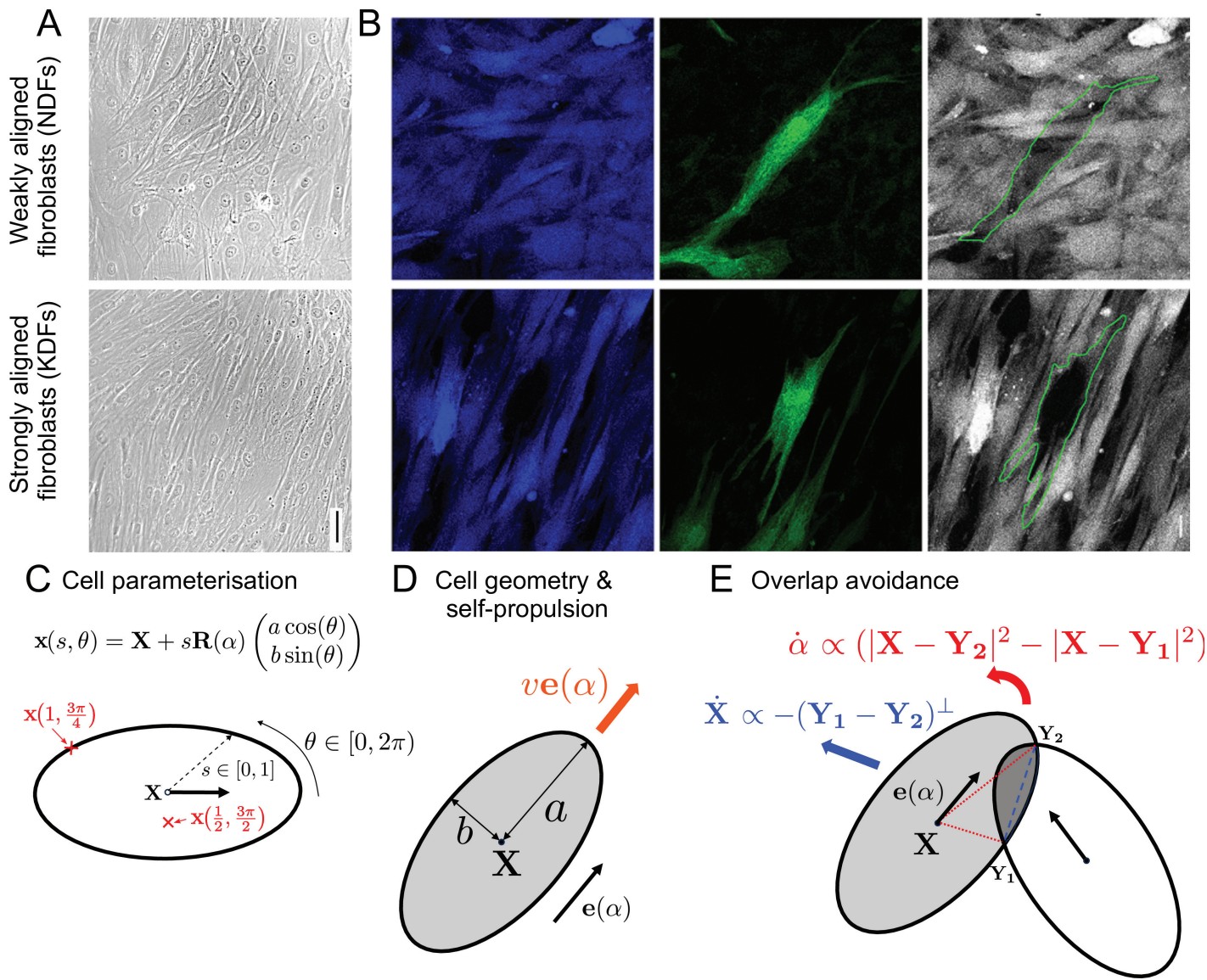

**Fig 1. A,B:** Experimental figures for weakly aligned normal dermal fibroblasts (NDFs), top row, and strongly aligned keloid derived fibroblasts (KDFs), bottom row. A: Phase microscopy pictures, scale bar 50 $\mu$m. B: Mosaically labelled cells with two different probes in order to investigate cell overlap (left: CellTrace Violet, middle: CellTrace Green, right: overlay), scale bar 20 $\mu$m. C,D,E: Model schematics. C: Ellipse parameterisation. D: Elliptic cell geometry of a single cell with centre $\mathbf{X}$, dimensions $a$ and $b$, orientation $\alpha$ and self-propulsion speed $w$. E: Effect of overlap avoidance upon one cell overlapping with another cell.

have investigated the difference in alignment behaviour of fibroblasts in healthy tissue (normal dermal fibroblasts, NDFs) as compared to dermofibroblasts in certain scar tissue (keloid derived fibroblasts, KDFs). KDFs were found to show stronger alignment over larger length scales, Fig 1A and 1B. Further we found that KDFs show less tendency to crawl on top of each other and form aligned supracellular actin bundles via cell-cell junctions, spanning multiple cells. Using mathematical modelling, we found in [55] that the increase in overlap avoidance can explain the stronger alignment. In this work we will use the differences found in NDFs and KDFs to motivate further model extensions. However, the model ingredients are applicable to other cells and situations and hence the findings are relevant beyond fibroblasts.

**Model ingredients.**   We build a mechanistic, agent-based model to describe the motion of individual cells interacting with neighbouring cells in 2D, where cells are approximated as ellipses. While typically elongated, real fibroblast shapes are of course more complex and include ruffles, and protruding and contracting lamellipodia [56, 57]. We argue that, over the timescale we are interested in, ellipses are a rough, but appropriate approximation of real cell shapes. We also choose ellipses, because they allow for a straight-forward description of dynamic cell shape changes (see Sec 3). Further our modelling approach can be extended to more complex cell shapes, which will be the subject of future work. In this work we do not model cell divisions, however, they could be included in a straight-forward manner.

The key model ingredients are

- *Environmental friction:* As usual in cell biology, we assume a friction-dominated regime. As a consequence, velocities (not accelerations) are proportional to forces. The strength of the friction with the substrate is given by $\eta$, which effectively sets a time scale.

- *Self-propulsion:* In the absence of interactions, cells move with fixed speed $w$ in the direction of their orientation. Orientational noise could be included in a straight forward manner, but is omitted in this work for the sake of simplicity.

- *Overlap avoidance:* When placed on a 2D substrate, many cells types will tend to avoid moving on top of other cells. The term *contact inhibition of locomotion* is sometimes used in this context. However, contact inhibition of locomotion more commonly refers to an active change of direction upon contact, as opposed to a more passive reaction, which is what we model here. Another commonly used term in this context is *repulsion*, however, in this work will use the term *overlap avoidance* to emphasise that the effect is short-ranged and driven by cell overlap. Note that "overlap" in 2D can be interpreted either as being positioned partly on top of each other, or allowing for some cell softness. We allow overlap avoidance to be tuneable, its strength given by parameter $\sigma$. If $\sigma = 0$, cells have no overlap avoidance, and for $\sigma \to \infty$, cells would behave as solid objects that never overlap/move on top of each other. To avoid overlap cells can

  - move to change their location,

  - turn to change their orientation, or

  - change their shape ($\to$ Sec 3).
    All these effects will be a consequence of the minimisation of a common energy term.

- *Cell-cell junctions:* In Sec 4 we model and investigate the effect of cell-cell junctions, where cells are elastically tethered to each other, which can affect their orientation and position.

- *Actin forces:* Also in Sec 4 we describe the presumed effect of supracellular actin cables that lead to cytoskeletal forces affecting cell orientation.

In this work we explore the effect of different parameters on the model without the inclusion of additional noise. However, we have added a short study on the effect of orientational noise in S1 Appendix, Sec 3 and in [55] we fitted the noise parameter to experimental data. In principle, other sources of stochasticity could be included in the model to e.g. explore randomly occurring protrusions.

## 2.2 Model derivation

We will show the derivation excluding cell deformability and cell-cell junctions. These will be considered in Sec 3 and Sec 4. More derivation details can be found in S1 Appendix, Sec 1 and

a summary of model parameter names and meaning can be found in S1 Appendix Tab. 1. We consider $N$ cells within the fixed domain $\Omega \in \mathbb{R}^2$, each with centroid position $\mathbf{X}_i = (X_i, Y_i) \in \mathbb{R}^2, i = 1, \ldots, N$ and orientation $\alpha_i \in [0, 2\pi), i = 1, \ldots, N$. Each cell is described by an ellipse with semi-major axis $a$ and semi-minor axis $b$ as shown in Fig 1D. The cell's area is given by $A = ab\pi$. In the absence of other cells, each cell self-propels with constant velocity $w$ in direction $\mathbf{e}(\alpha_i) = (\cos(\alpha_i), \sin(\alpha_i))^T$, where superscript $T$ denotes the transpose. We assume that this self propulsion encompasses the mechanism of an individual cell moving in a directed way across the substrate. An alternative approach would be to model cell motion as a persistent random walk, as has been done in [58, 59].

We derive the governing equations using energy minimisation. An alternative model derivation based on force balance, which leads to the same governing equations, can be found in S1 Appendix, Sec 1. Focusing on one cell positioned at $\mathbf{X}(t)$ with orientation $\alpha(t)$ at time $t$, we parameterise the points inside the cell, $\mathbf{x}$, by

$$\mathbf{x}(t, s, \theta) = \mathbf{X}(t) + s\mathbf{R}(\alpha(t))\mathbf{k}(\theta), \qquad s \in [0, 1], \ \theta \in [0, 2\pi), \tag{1}$$

where $s$ encodes the distance from the centre of the cell $\mathbf{X}$ to the point $\mathbf{x}$ as a proportion of the distance from the centre of the cell to the boundary and $\theta$ encodes the angle parameter, see Fig 1C. The rotation matrix $\mathbf{R}(\alpha)$ and the shape vector $\mathbf{k}(\theta)$ are defined by

$$\mathbf{R}(\alpha) = \begin{pmatrix} \cos(\alpha) & -\sin(\alpha) \\ \sin(\alpha) & \cos(\alpha) \end{pmatrix} \quad \text{and} \quad \mathbf{k}(\theta) = \begin{pmatrix} a\cos(\theta) \\ b\sin(\theta) \end{pmatrix}.$$

We assume that at every time step $\Delta t$, the system minimizes a total energy $E_{\text{tot}}$, which, for the base model, is the sum of contributions from friction $E_{\text{friction}}$, from overlap avoidance $E_{\text{overlap}}$ and from self-propulsion $E_{\text{prop}}$. All terms inside the integrals below represent the effect of each contribution on one point $\mathbf{x}$ inside the ellipse. We then obtain the total energy for one cell by integrating over whole (elliptic) cell area, i.e. with respect to $s$ and $\theta$. The chosen parametrisation given in (1) leads to the appearance of the area element $abs$ (which accounts for the fact that closer to the cell centre one step in $s$-direction contributes less area) in the integrals in (2), (3) and (4). $E_{\text{friction}}$ models friction with the environment by comparing how much points have moved between time $t$ and time $t - \Delta t$:

$$E_{\text{friction}} = \eta \int_0^{2\pi} \int_0^1 abs \frac{|\mathbf{x}(t, s, \theta) - \mathbf{x}(t - \Delta t, s, \theta)|^2}{2\Delta t} \, ds \, d\theta. \tag{2}$$

The overlap avoidance term $E_{\text{overlap}}$ is modelled by an energy potential $V$ which includes overlap avoidance interactions with all other cells. The choice of $V$ will be discussed below.

$$E_{\text{overlap}} = \int_0^{2\pi} \int_0^1 abs V(\mathbf{x}(t, s, \theta)) \, ds \, d\theta. \tag{3}$$

Finally, we want to model self-propulsion. One way to do this is to include it in the energy formulation by prescribing a force $\mathbf{F}$ acting on the cell. In the course of the derivation it is chosen to be $\mathbf{F} = w\eta\mathbf{e}(\alpha)$, i.e. acting in the direction of the orientation and proportional to the experienced friction $\eta$. This choice of $\mathbf{F}$ leads to a self-propulsion speed that is independent of friction (choosing $\mathbf{F}$ not proportional to $\eta$ would only lead to a different definition of the non-dimensional quantities below).

$$E_{\text{prop}} = -\int_0^{2\pi} \int_0^1 abs\mathbf{F} \cdot \mathbf{x}(t, s, \theta) \, ds \, d\theta. \tag{4}$$

The total energy is then given by summing (2), (3) and (4)

$$E_{\text{tot}} = E_{\text{friction}} + E_{\text{overlap}} + E_{\text{prop}}. \tag{5}$$

We obtain governing equations by minimising this energy in each time step. In other words, in each time step the cell can change its characteristics (position, orientation, shape) to decrease the energy. Calculation details can be found in S1 Appendix Sec 1. The main derivation steps for the base model are: 1. Differentiation with respect to $\mathbf{X}$ and $\alpha$ respectively (treating all other variables in the energy potential as constants). 2. Setting the derivative to zero and taking the limit $\Delta t \to 0$. 3. Evaluation of the integrals. We then obtain the following differential equations for the motion of one cell

$$\frac{\mathrm{d}\mathbf{X}}{\mathrm{d}t} = -\frac{1}{\eta\pi} \int_0^{2\pi} \int_0^1 s\nabla V \, \mathrm{d}s \, \mathrm{d}\theta + w\mathbf{e}(\alpha), \tag{6a}$$

$$\frac{\mathrm{d}\alpha}{\mathrm{d}t} = -\frac{4}{\eta\pi(a^2+b^2)} \int_0^{2\pi} \int_0^1 s^2 \nabla V \cdot (\mathbf{R}\mathbf{k}(\theta))^\perp \, \mathrm{d}s \, \mathrm{d}\theta. \tag{6b}$$

The superscript $\perp$ describes the left-turned normal vector. The two equations in (6) show how the position and orientations are influenced by the force and torque associated with $V$ respectively. Note that we are working in a friction-dominated regime, which is why velocity and angular velocity (as opposed to acceleration and angular acceleration) are proportional to force and torque.

**Choice of overlap potential $V$.** The potential $V$ describes the influence of overlap, where $V > 0$ describes overlap avoidance and $V < 0$ overlap preference. Many choices of $V$ are possible: e.g. since cells might be thicker closer to the cell center, overlap closer to the cell center could be punished more than further away. However, due to the governing equations in (6) being formulated in terms of integrals of the gradient of the potential $V$, complicated shapes of $V$ are computationally harder to evaluate, especially in the context of collective dynamics when this will need to be computed numerous times at each time step. We therefore choose $V$ to be constant with value $\sigma$ in regions of overlap and zero elsewhere: For two overlapping ellipses with domains $\mathcal{A}$ and $\mathcal{B}$ we define $V(\mathbf{x}) = \sigma \mathbb{1}_{\mathcal{A}\cap\mathcal{B}}(\mathbf{x})$, where $\mathbb{1}_{\mathcal{A}\cap\mathcal{B}}(\mathbf{x})$ is the indicator function which equals 1 if $\mathbf{x} \in \mathcal{A} \cap \mathcal{B}$ and 0 otherwise. The strength of this potential is $\sigma \in \mathbb{R}$. If $\sigma > 0$, the cells experience repulsion in response to overlap, and if $\sigma < 0$, the cells experience attraction. In this work $\sigma > 0$. As a result of this choice of potential $V$, two cells only experience overlap avoidance upon overlapping with each other, hence we define $\mathcal{N}_i$ as the set of indices of cells that overlap with the $i$-th cell.

**Final base model.** We non-dimensionalise the model using as reference time $\frac{A\eta}{\sigma}$, as reference length $\sqrt{\frac{A}{\pi}}$ and define $r = a/b$ as the cell's aspect ratio. The above choice of $V$ allows to evaluate the integrals in (6) explicitly (for calculation details see S1 Appendix, Sec 1). The resulting equations can be formulated such that they depend only on the points of overlap between cells $i$ and $j$, denoted by $\mathbf{Y}_k^{ij}$, where up to $k = 4$ points of overlap are possible. This is computationally advantageous since only the points of overlap need to be found, instead of areas of overlap which would be more computationally costly. In the following $K_{ij} = 1$ or $K_{ij} = 2$ denotes the number of overlap point pairs between cell $i$ and cell $j$ (having one or three points of overlap can be reduced to having zero or two points of overlap). The overlap points are ordered such that they traverse the boundary of the cell in an anti-clockwise direction. $\mathbf{Y}_1^{ij}$ and $\mathbf{Y}_2^{ij}$ (and $\mathbf{Y}_3^{ij}$ and $\mathbf{Y}_4^{ij}$) are chosen in such a way that the boundary segment of cell $i$ between

these pairs of points is contained in the domain of cell $j$, see Fig 1D.

$$\frac{\mathrm{d}\mathbf{X}_i}{\mathrm{d}t} = -\sum_{j \in \mathcal{N}_i} \sum_{k=1}^{K_{ij}} (\mathbf{Y}_{2k-1}^{ij} - \mathbf{Y}_{2k}^{ij})^\perp + v\mathbf{e}(\alpha_i), \tag{7a}$$

$$\frac{\mathrm{d}\alpha_i}{\mathrm{d}t} = \frac{2r}{r^2+1} \sum_{j \in \mathcal{N}_i} \sum_{k=1}^{K_{ij}} (|\mathbf{X_i} - \mathbf{Y}_{2k}^{ij}|^2 - |\mathbf{X_i} - \mathbf{Y}_{2k-1}^{ij}|^2), \tag{7b}$$

where the non-dimensional quantity $v$ is given by $v = \frac{w\eta}{\sigma}\sqrt{\pi A}$ and can be interpreted as comparing the strength of repulsion in the presence of friction to the self-propulsion speed (see more interpretation in the results section below). These two governing equations are supplemented with initial conditions and boundary conditions. Throughout this work the domain is a square box with side length $L$ and cells are initially placed randomly inside the box with a random orientation. Further, we use periodic boundary conditions.

**Interpretation for two cells.** To understand the equations better, we consider a situation where there is only interaction between one cell with center $\mathbf{X}$ and orientation $\alpha$, and one other cell. If there is only one pair of overlap points, $\mathbf{Y}_1$ and $\mathbf{Y}_2$, then (7) reduces to

$$\frac{\mathrm{d}\mathbf{X}}{\mathrm{d}t} = -(\mathbf{Y}_1 - \mathbf{Y}_2)^\perp + v\mathbf{e}(\alpha), \tag{8a}$$

$$\frac{\mathrm{d}\alpha}{\mathrm{d}t} = \frac{2r}{r^2+1}(|\mathbf{X} - \mathbf{Y}_2|^2 - |\mathbf{X} - \mathbf{Y}_1|^2). \tag{8b}$$

We see in (8a) that the cell's center is being pushed in the direction normal to the vector connecting the points of overlap. Further, we see in (8b) that the change in orientation depends on the difference in lengths of the segments connecting the cell center with the intersection points, turning the cell in the direction from the shorter to the longer one, see Fig 1D. This shows that cells will both move away from each other, and reorient themselves in order to minimise cell overlap. These are both behaviours that can be observed in experimental videos, see [55]. Compared to a more *ad hoc model* of simple repulsion between centroids there are two main differences: Firstly the direction of movement of the centroids would be different: movement would occur along the line connecting the centroids as opposed to $-(\mathbf{Y}_1 - \mathbf{Y}_2)^\perp$. Likely this difference would still lead to a similar behaviour in terms of cell populations spreading out and forming a monolayer. However, secondly (and more importantly), our model also provides an equation for how the cell orientation changes without the need of extra assumptions.

## 2.3 Results 1: The base model

Computational details can be found in S1 Appendix, Sec 2. We have made some qualitative comparisons of the model with experimental data in [55]. In this paper, we focus on the computational results of the model.

**Alignment increases over time and for larger aspect ratios.** We start by demonstrating basic model behaviour. In Fig 2A and 2B we see the typical behaviour of the base model with overlap avoidance. Over time cell overlap decreases and alignment increases until a dynamic equilibrium is reached. At this point cells still move, but the alignment parameter stays relatively constant. Further, we observe that cells tend to be aligned with their direct neighbours, but this alignment is local and doesn't typically go beyond one or two cell lengths. This is related to packing problems, where one studies how and how densely objects of a certain shape can be placed in space without overlapping. Such problems are highly non-trivial, but

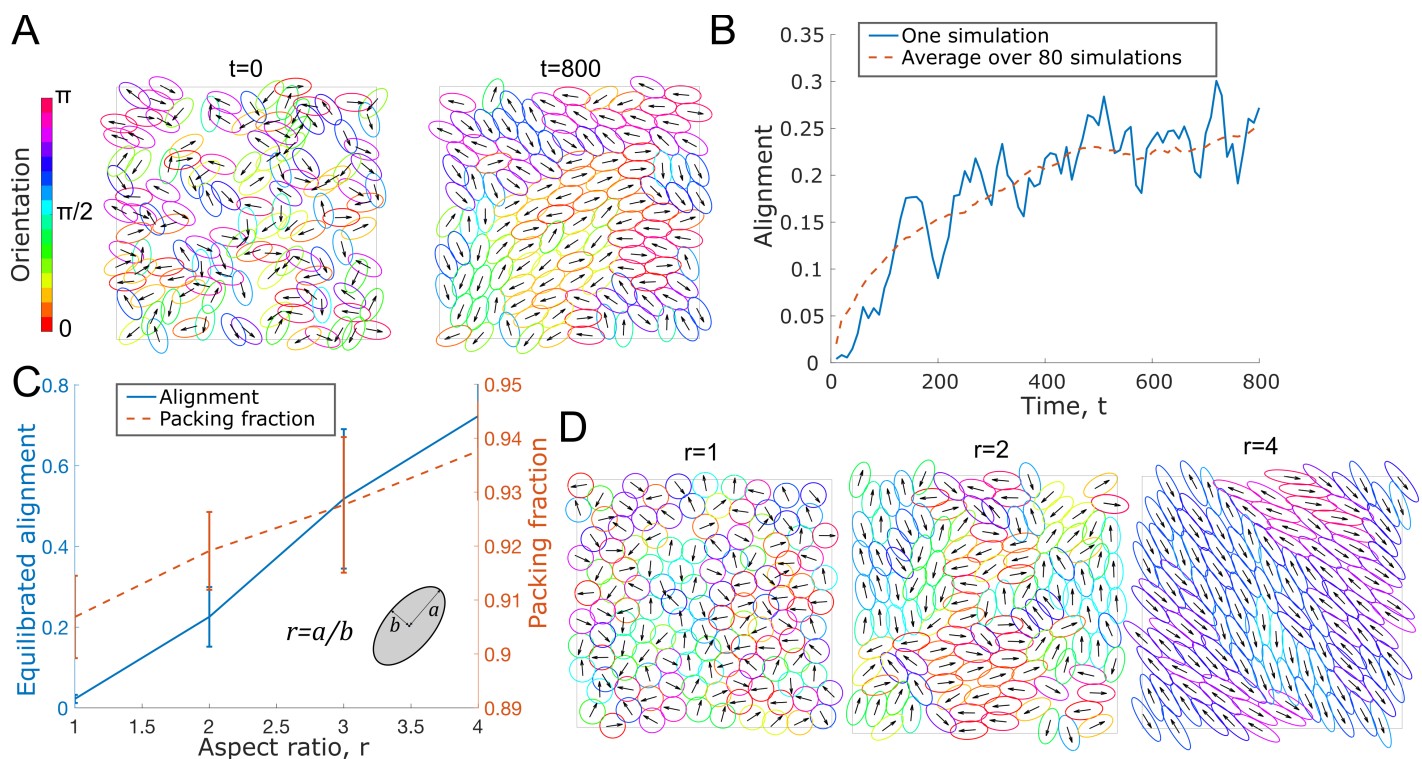

**Fig 2.** A: Simulation snapshots at time $t = 0$ and $t = 800$ showing cells for an example simulation, color indicates nematic orientation, arrows indicate orientation. See also S1 Video. B: Alignment parameter over time as defined in S1 Appendix, Eq (1) shown for one individual simulation (blue, solid) and averaged over 80 simulations (orange, dashed). Parameters for A,B: $v = 0.2$, $N = 125$, $L = 20$, $r = 2$. C: Alignment parameter and packing fraction (see S1 Appendix, Sec 2), measured at the dynamic equilibrium, plotted against the cell aspect ratio $r$, averaged over 60 simulations, error bars represent standard deviation. D: Simulation snapshot at the (final) time points for three different aspect ratios. Parameters for C,D: $v = 0.5$, $N = 125$, $L = 20$.

well-studied for non-moving, completely solid particles, and for symmetric and elongated particles in 2D and 3D (see e.g. [8–10]). Next we inspect how the aspect ratio impacts alignment. We find that increasing the aspect ratio $r$ of the cells leads to increased alignment. This can be seen in Fig 2C and 2D. This is a result that can also be found in [18]. It is similarly found experimentally in [60] and computationally in [61] that the ability of bacteria to swarm effectively is modulated by the cell aspect ratio. Another model that shows that aspect ratio increases alignment can be found in [62]. Investigating this further, we see in Fig 2C that increasing the aspect ratio of the cells also leads to a higher packing fraction, meaning that there is less cell overlap in the population. Interestingly, for non-moving, solid ellipses [10] found a similar dependence of the alignment parameter on the aspect ratio, albeit with a peak near $r \approx 1.3$.

**Optimal ratio of cell speed to overlap avoidance for alignment.** The non-dimensional parameter $v$ is proportional to the self-propulsion speed divided by the strength of overlap avoidance, $v \propto \frac{w}{\sigma}$, and can be interpreted as the ratio between two time scales $t_1/t_2$, where $t_1 = \frac{A\eta}{\sigma}$ is the time scale of the movement caused by overlap avoidance acting against friction. The time scale $t_2 = w/\sqrt{A/\pi}$ is the time it takes a self-propelling cell of speed $w$ to move one reference length $\sqrt{A/\pi}$. Varying $v$ and measuring the resulting alignment parameter at the dynamic equilibrium, we find a non-monotone dependence on $v$, with a maximal alignment at $v = 0.2$, see Fig 3A and 3E. We hypothesised that if there is too little self-propulsion, overlap avoidance pushes cells into a little or no overlap configuration, after which cells do not move

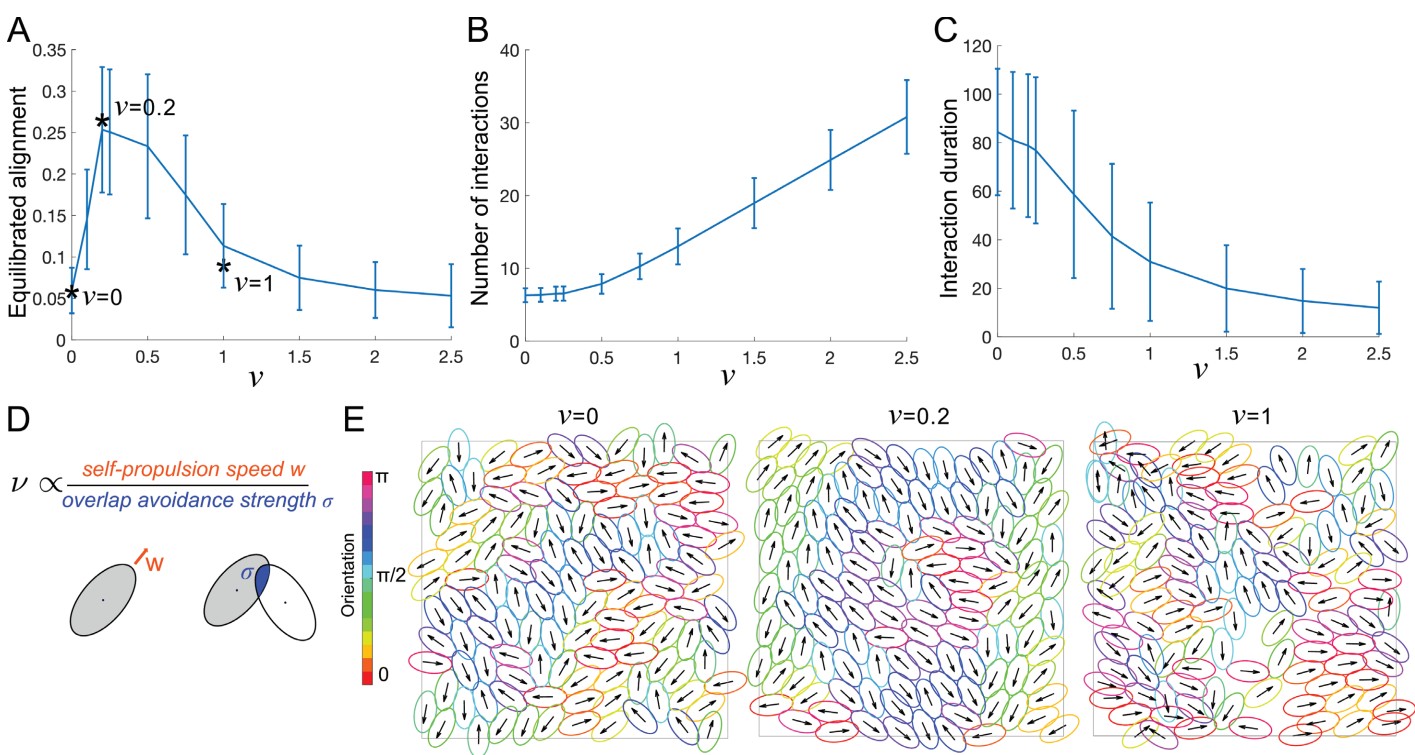

**Fig 3.** A: Alignment parameter measured at the dynamic equilibrium as a function of $v$, averaged over 80 simulations, error bars show standard deviation, stars mark simulations in E. B,C: Number of interaction partners per cell (B) and duration of each pair-wise cell interaction (C) between $t = 0$ and $t = 10$ plotted against $v$, averaged over all 125 cells and 5 simulations. Error bars show standard deviation. D: Schematic explanation of $v$. E: Simulation snapshots at final time for $v$-values marked with a star in A. Colors and arrows as in Fig 2. Fixed parameters: $r = 2$, $N = 125$, $L = 20$.

much and the alignment doesn't increase further. In that situation, pairs of cells might therefore interact with each other for a long duration, but each cell doesn't interact with a large number of cells. Self-propulsion, on the other hand, might lead to a re-shuffling of cell contacts, leading to shorter-lived, but more numerous interactions. To test this, we quantified the number of cell contacts over 100 time points for a duration of t = 10, and the typical interaction duration (see S1 Appendix, Sec 2 for details on the quantification). Indeed, we found that the number of interaction partners increases with $v$ (Fig 3B) and that the interaction duration decreases with $v$ (Fig 3C). This leads us to suggest the following explanation for the optimal value $v$ found in Fig 3A: Effective alignment requires cells: 1. To be in contact with their neighbours over a sufficiently long duration (for overlap avoidance to take effect and cause cells to re-orient locally, time scale given by $t_1 = \frac{A\eta}{\sigma}$) and 2. To be in contact with sufficiently numerous different cells (time scale given by $t_2 = w/\sqrt{A/\pi}$) in order for the local order to be propagated beyond immediate neighbours. This finding underlines an important difference in alignment dynamics between active, self-propelling matter, and passive, non self-propelling matter: Self-propulsion will generally increase the number of different neighbours a given cell will interact with, while without self-propulsion there will be fewer interaction partners, but potentially longer interaction times.

**Results are consistent with similar models.** In relation to the field of polar active matter (see e.g. review paper [13] and [4, 18, 63–65] for applications to biology) our model can be thought of as a 'dry' nematic system, which means that momentum is not conserved and that

the interactions do not have a polar preference. The latter being a consequence of the elongated shape of the particles and that the interactions arise through overlap avoidance. Several of our results are consistent with simulations of self-propelled hard rods [66–68], such as the appearance of nematically ordered regions. In [66] it was noted that self-propulsion enhances nematic order, which is also what we observe for small propulsion speeds, however in our model self-propulsion can also hinder large scale alignment. This might be a consequence of our particles being "soft", i.e. that overlap is merely punished, not forbidden. In [69], where soft interactions are modelled, it is found that increasing self propulsion speed causes cells to transition from a solid like state to a liquid like state. This can be compared to our results by noting that cells do indeed become jammed for very low values of $v$ in which we see little alignment. As $v$ increases the cells are able to move around more and hence behave more like a liquid. Also other works have analysed phase transitions between ordered and unordered state occurring at a critical density which depends on the particle size, noise and self-propulsion speed (see e.g. [66, 70]). While this was not the focus of this work (we work at high densities and with no noise), we anticipate that our model would show the same kind of behaviour. We also expect similar behaviour with regards to orientational defects. As a future direction, it would be interesting to investigate whether our model replicates general alignment properties of continuum models of active nematic systems. These tend to be developed from hydrodynamic theory, as reviewed in [13], and can be useful to analyse the overarching macroscopic properties of these systems (such as the stability of steady states), without going into details on a cell level.

## 3 Modelling shape changes

### 3.1 Biological motivation

Cells vary in their ability to change shape. Many bacteria, for example, are surrounded by a stiff cell wall, leading to few shape deformations. Most other cell types, including fibroblasts, are soft and deformable, and change their shape dynamically due to internal changes, or in reaction to their surroundings. For example, they can get squished together when confined or become elongated when attached. While cells can also change shape in the absence of other cells, here we only consider shape changes in reaction to interactions with other cells.

### 3.2 Model derivation

We investigate the effect of allowing cells to dynamically change their shape in response to overlap. A summary of model parameter names and meaning can be found in S1 Appendix, Tab. 1. We restrict allowed cell shapes to changing the aspect ratio $r = a/b$, where $r(t)$ is now a function of $t$, while maintaining a constant cell area $A$. To avoid unrealistically large (or small) aspect ratios, we add a term to the energy that punishes deviations from some preferred aspect ratio $\bar{r}$, which we set to $\bar{r} = 2$. In other words, in the absence of interactions, cell shape will relax towards having the preferred aspect ratio. The strength of this relaxation is given by $g$. This adds an extra term $E_{\text{shape}}$ to the $E_{\text{tot}}$ of the base model given in (5),

$$E_{\text{shape}} = \frac{g}{2}\left[(r - \bar{r})^2 + \left(\frac{1}{r} - \frac{1}{\bar{r}}\right)^2\right].$$

Note that $E_{\text{shape}}$ is symmetric with respect to $r \to \frac{1}{r}, \bar{r} \to \frac{1}{\bar{r}}$. This is to ensure that the shape relaxation behaviour is the same along both axes. The model derivation now follows the same steps as before, for some more details see S1 Appendix, Sec 1. The equations for $\mathbf{X}$ and $\alpha$ remain unchanged by this. For one cell, we obtain the following equation for how the aspect

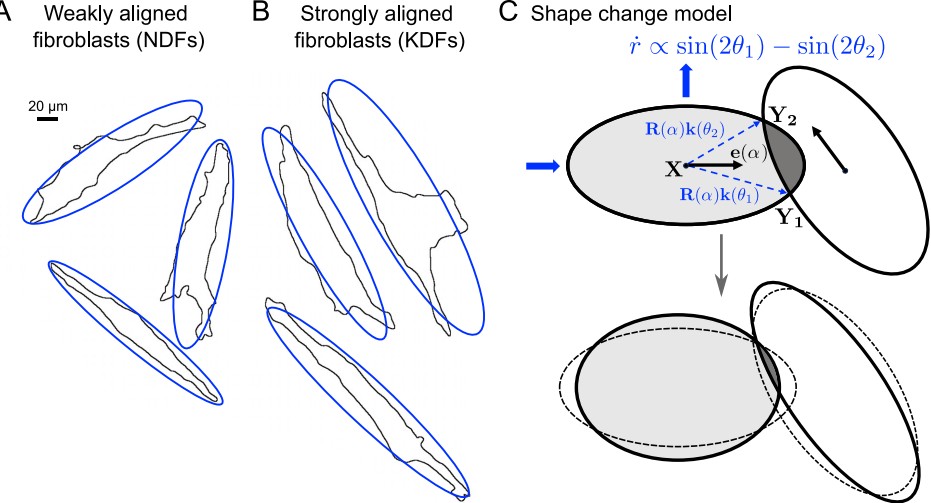

**Fig 4.** A,B: Single weakly aligned fibroblast (A) and strongly aligned fibroblast (B) cells show a range of different shapes in confluent cultures. No difference can be observed between the two samples [55]. C: Schematic of effect of shape change model.

ratio $r$ changes over time as a result of overlap avoidance and shape relaxation:

$$\dot{r} = -\frac{16r^3}{A\eta(r^2+1)}\int_0^{2\pi}\int_0^1 s^2 \nabla V \cdot \left(\mathbf{R}(\alpha)\frac{d\mathbf{k}}{dr}\right)ds\,d\theta + \frac{16\pi g(1+\bar{r}r^3)}{A^2\eta(1+r^2)}\left(1-\frac{r}{\bar{r}}\right).$$

Using the same overlap potential as defined above, we obtain (written in non-dimensional form)

$$\dot{r}_i = \frac{4r^2}{r^2+1}\sum_{j\in\mathcal{N}_i}\sum_{k=1}^{K_{ij}}[\sin(2\theta_{2k-1}^{ij})-\sin(2\theta_{2k}^{ij})] + 16\gamma\frac{1+\bar{r}r^3}{1+r^2}\left(1-\frac{r}{\bar{r}}\right),$$

where $\gamma = \frac{\pi g}{A\sigma}$ and $\theta_k^{ij}$ corresponds to the $\theta$ value that parameterises the overlap point $\mathbf{Y}_k^{ij} = \mathbf{X}_i + \mathbf{R}(\alpha_i)\mathbf{k}(\theta_k^{ij})$.

**Interpretation for two cells.** In the case of the interaction between only two cells with two overlap points (as in Fig 4C) we have

$$\dot{r} = \frac{4r^2}{r^2+1}[\sin(2\theta_1)-\sin(2\theta_2)] + 16\gamma\frac{1+\bar{r}r^3}{1+r^2}\left(1-\frac{r}{\bar{r}}\right).$$

If, as in Fig 4C, we have that $\theta_1 \in \left(-\frac{\pi}{2}, 0\right)$ and $\theta_2 \in \left(0, \frac{\pi}{2}\right)$, then $\sin(2\theta_1)-\sin(2\theta_2) < 0$ and the first term will cause the aspect ratio $r$ to decrease. This is a result of the cell shortening to avoid overlap. The second term will always act to restore the aspect ratio towards $\bar{r}$. Note that there is only one additional parameter, $\gamma$, quantifying the restoring force, but no extra parameter quantifying the initial deformation. The reason for this is that shape changes are also driven by minimising overlap and hence based on the same energy term, $E_{\text{overlap}}$, as the other dynamics (turning and non-propulsion driven translations) driven by overlap avoidance.

### 3.3 Results 2: The base model with shape changes

**Deformable cells lead to increased alignment.**   We investigate the effect of $\gamma$, the strength of the shape restoring force. Small $\gamma$ means cells are more easily deformable, while the limit $\gamma \to \infty$ corresponds to non-deformable cells with fixed aspect ratio $\bar{r}$ (corresponding to the base model discussed in Sec 2). Fig 5A shows that higher deformability correlates with more alignment, suggesting that changes in aspect ratio aid alignment. Further, we found that allowing cells to deform more increases the average aspect ratio in the population, Fig 5B. We also assessed whether the optimal value of $v$ found in Fig 3A is affected by deformability. Fig 5C shows that indeed, maximal alignment is now reached for larger values of $v$, i.e. bigger self-propulsion speeds or smaller overlap avoidance.

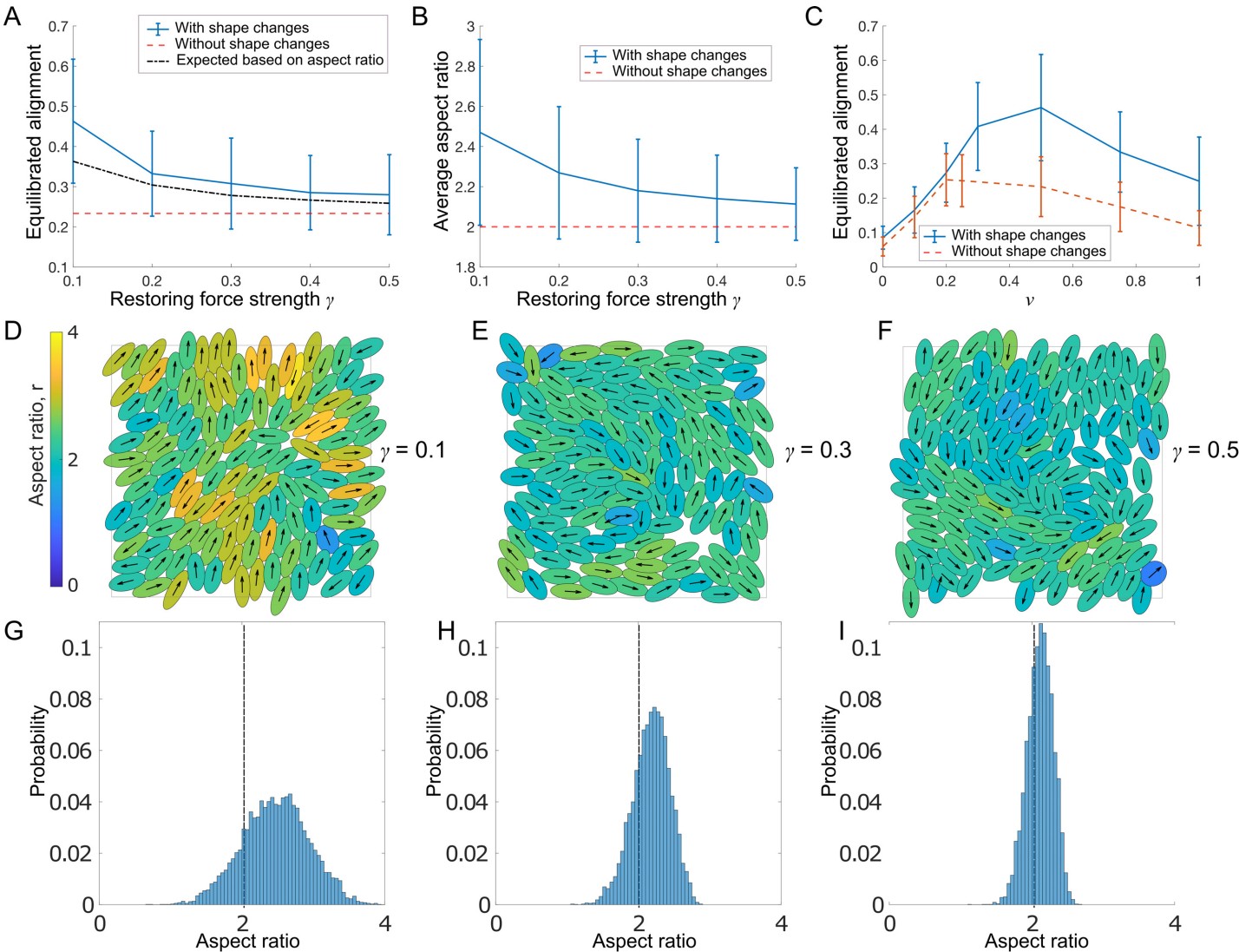

**Fig 5.** A,B: Equilibrated alignment (A) and average aspect ratio (B) measured for different values of $\gamma$ and compared to non-deformable cells with $v = 0.5$. In A: Black, dash-dotted line shows expected alignment based on the measured average aspect ratio (see text). C: Equilibrated alignment for different values of $v$ for deformable and non-deformable cells with $\gamma = 0.1$. D-F: Simulation snapshots at equilibrium for three different values of $\gamma$. See also S2 Video. Colors indicate aspect ratio. G-I: Distribution of aspect ratio measured for $\gamma$-values in D-F using values pooled from 60 simulations at equilibrium. Fixed parameters: $N = 125$, $L = 20$, $\bar{r} = 2$.

## 4 Modelling cell-cell junctions and actin forces

### 4.1 Biological motivation

In some types of cells, neighbouring cells can form cell-cell junctions. These provide a mechanical coupling as well as a way for cells to exchange signals, i.e. communicate. We model two potential effects of cell-cell junctions: 1. Elastic, reversible connections between two points on the edges of neighbouring cells. 2. Formation of supracellular actin bundles that lead to a bending force. The actin cytoskeleton is the main force generator for moving cells and plays a major role in determining cell shape and polarity. In [55] we observed that the actin network within a cell shares the orientation of the cell itself, with the major bundles typically running along the long axis. Further we found that for keloid fibroblasts (KDFs), neighbouring cells can form supracellular actin bundles, i.e. that the actin bundles of neighbouring cells visually appear to be connected to each other in a smooth manner, without an abrupt change in direction of the actin bundles at the cell-cell junction. This suggests a potential mechanical linkage mediated by cell-cell junctions that causes cells to align their cytoskeleton, and therefore themselves. We will focus on the latter, including elastic connections as a means for these supracellular actin bundles to form. Since the actin bundles are predominantly found to run along the length of each cell, we will only consider cell-cell junctions forming at the front and back of each cell, but see S1 Appendix Sec 3 for some results on cell-cell junctions forming at the sides.

### 4.2 Model derivation

A summary of model parameter names and meaning can be found in S1 Appendix, Tab. 1.

**Elastic connections.** In principle cell-cell junctions could form whenever two points in the domain of the ellipses are in close contact. However, since we are interested in cell-cell junctions as a way to mediate the formation of supracellular actin bundles running parallel to the long axis of the cell, we will focus on cell-cell junctions at the front and the rear of cells. We therefore restrict ourselves to only allowing front-to-back, front-to-front and back-to-back junctions. We model these connections as Hookean springs with rest length zero. The effect of cell-cell junctions at different locations around the cell boundary has been explored in S1 Appendix Sec 3, where we allowed cell-cell junctions to also form at two points at the cell sides. In principle such connections could be created and broken stochastically, with a distance (or force) dependent breakage rate. However these processes likely happen on a much faster timescale than the overall alignment dynamics. Such a stochastic formulation would also introduce additional parameters, the values of which are hard to infer from experimental data. Further these additional unknown parameters would significantly complicate the analysis of the model. We therefore assume deterministic springs that form and stay in place whenever the distance between connection points are below some critical distance, and break when stretched beyond that distance.

**Trans-cellular actin cables.** Detailed models of actin bundles within cells have been developed e.g. in [71, 72]. Here we are only interested in the potential effect of trans-cellular actin cables on the orientation of neighbouring cells. We therefore don't provide a more detailed model of the dynamics of the actin bundle *within* each cell, but rather focus on the potential effect of the *connected* actin cable. Inside the cells, we just approximate each bundle as a straight inextensible rod. Motivated by the biological findings in [55], summarised above, we model the trans-cellular actin cable as an inextensible rod with a given bending stiffness. For a general rod discretised with a uniform step length $q$, resulting in the points $\mathbf{x}_i \in \mathbb{R}^2$, $i = 1$,

..., $K$, the bending energy of strength $m$ is given by

$$\frac{m}{2}\sum_{i=2}^{K-1}\left(\frac{|\mathbf{x}_{i-1}-2\mathbf{x}_i+\mathbf{x}_{i+1}|}{q^2}\right)^2 q.$$

Note that its continuous counterpart would have an integral instead of the sum and the norm of the second derivative instead of the quotient. We discretise the supracellular actin bundle using three points, the two end points not involved in the cell junction, plus the midpoint of the cell-cell junction points. Further we use $q \approx 2a$. This formulation has the advantage that it leads to a very simple bending energy. For example, if the front of a cell positioned at $\mathbf{X}_1$ with orientation $\alpha_1$ is connected to the rear of a cell positioned at $\mathbf{X}_2$ with orientation $\alpha_2$, the corresponding bending energy would be

$$\frac{m}{2}\left(\frac{|\mathbf{X}_1 - a\mathbf{e}(\alpha_1) - 2\frac{\mathbf{X}_1 + a\mathbf{e}(\alpha_1) + \mathbf{X}_2 - a\mathbf{e}(\alpha_2)}{2} + \mathbf{X}_2 + a\mathbf{e}(\alpha_2)|}{(2a)^2}\right)^2 2a = \frac{m}{4a}|\mathbf{e}(\alpha_1) - \mathbf{e}(\alpha_2)|^2.$$

This shows that the bending will only affect the cells' orientation, not their positions.

**Incorporation into the full model.** We denote the strength of the Hookean springs describing the cell-cell junctions by $k$ and assume junctions will exist whenever potential connection points are within distance $l$ of each other. To distinguish between front and back connections we define the front and back ends of a cell by $\mathbf{X}^\pm := \mathbf{X} \pm a\mathbf{e}(\alpha)$, and the two relevant index sets by

$$\mathcal{N}^{\text{fb}} = \{j \in \{1,..N\} \mid |\mathbf{X}^\mp - \mathbf{X}_j^\pm| < l\}, \quad \mathcal{N}^{\text{ff,bb}} = \{j \in \{1,..N\} \mid |\mathbf{X}^\pm - \mathbf{X}_j^\pm| < l\}. \tag{9}$$

The set $\mathcal{N}^{\text{fb}}$ describes front-to-back junctions and $\mathcal{N}^{\text{ff,bb}}$ describes front-to-front and back-to-back junctions. The new contribution to the total energy for a cell positioned at $\mathbf{X}$ with orientation $\alpha$ now takes the form

$$E_{\text{junction}} = \sum_{j \in \mathcal{N}^{\text{fb}}}\left[\frac{k}{2}|\mathbf{X}^\pm - \mathbf{X}_j^\mp|^2 + \frac{m}{4a}|\mathbf{e}(\alpha) - \mathbf{e}(\alpha_j)|^2\right]$$

$$+ \sum_{j \in \mathcal{N}^{\text{ff,bb}}}\left[\frac{k}{2}|\mathbf{X}^\pm - \mathbf{X}_j^\pm|^2 + \frac{m}{4a}|\mathbf{e}(\alpha) + \mathbf{e}(\alpha_j)|^2\right].$$

The model derivation now follows the same steps as described in Sec 2 and yields, in non-dimensional form,

$$\frac{d\mathbf{X}_i}{dt} = -\sum_{j \in \mathcal{N}_i}\sum_{k=1}^{K_{ij}}(\mathbf{Y}_{2k-1}^{ij} - \mathbf{Y}_{2k}^{ij})^\perp + v\mathbf{e}(\alpha) + \kappa\left[\sum_{j \in \mathcal{N}_i^{\text{fb}}}(\mathbf{X}_j^\mp - \mathbf{X}_i^\pm) + \sum_{j \in \mathcal{N}_i^{\text{ff,bb}}}(\mathbf{X}_j^\pm - \mathbf{X}_i^\pm)\right],$$

$$\frac{d\alpha_i}{dt} = \frac{2r}{r^2+1}\sum_{j \in \mathcal{N}_i}\sum_{k=1}^{K_{ij}}(|\mathbf{X_i} - \mathbf{Y}_{2k}^{ij}|^2 - |\mathbf{X_i} - \mathbf{Y}_{2k-1}^{ij}|^2)$$

$$+ 4\kappa\frac{r^{3/2}}{r^2+1}\left[\sum_{j \in \mathcal{N}_i^{\text{fb}}}(\mathbf{X}_j^\mp - \mathbf{X}_i^\pm) \cdot (\pm\mathbf{e}^\perp(\alpha_i)) + \sum_{j \in \mathcal{N}_i^{\text{ff,bb}}}(\mathbf{X}_j^\pm - \mathbf{X}_i^\pm) \cdot (\pm\mathbf{e}^\perp(\alpha_i))\right]$$

$$+ \mu\frac{\sqrt{r}}{r^2+1}\left[-\sum_{j \in \mathcal{N}_i^{\text{fb}}}\sin(\alpha_i - \alpha_j) + \sum_{j \in \mathcal{N}_i^{\text{ff,bb}}}\sin(\alpha_i - \alpha_j)\right],$$

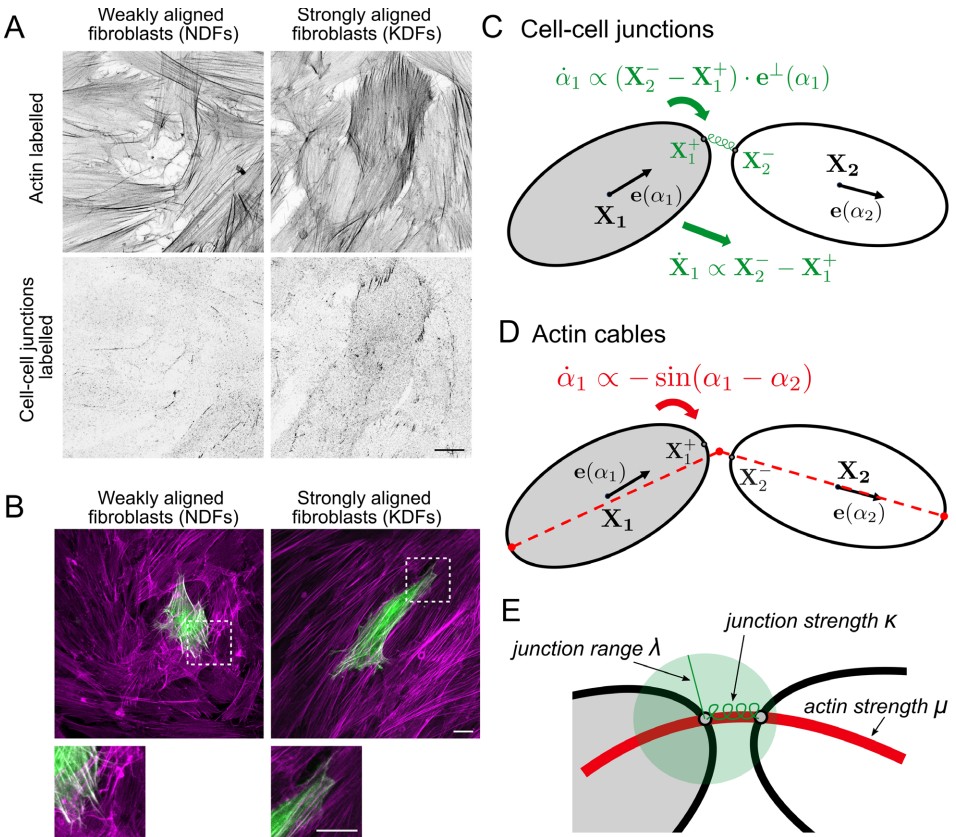

**Fig 6.** A: NDFs and KDFs stained for F-actin and N-cadherin (a marker for a type of cell-cell junction), revealing an enhanced localisation of N-cadherin at cell-cell junctions in KDF. Scale bar 20 $\mu m$. B: Supracellular actin bundles. NDF and KDF cultured in vitro for 48 hours, mosaically transfected with EGFP-LifeAct (green) and stained for F-actin (magenta). KDF display more aligned actin bundles spanning multiple cells. Scale bar 20 $\mu m$. C,D: Schematics showing how the cell-cell junctions (C) and actin forces (D) affect changes in cell position and orientation. E: Explanation of new non-dimensional parameters $\lambda$, $\kappa$ and $\mu$.

where the index sets $\mathcal{N}^{\text{fb}}$ and $\mathcal{N}^{\text{ff,bb}}$ are as defined in (9) with $l$ replaced by the non-dimensional $\lambda = \frac{l\sqrt{\pi}}{\sqrt{A}}$. Further we have defined the two non-dimensional quantities $\kappa = \frac{k}{\sigma}$ and $\mu = \frac{2m}{\sigma}\left(\frac{\pi}{A}\right)^{3/2}$, which compare the junction strength and bending strength to the strength of overlap avoidance. We assume that cell aspect ratio, $r$, is constant.

**Interpretation for two cells.** To understand the effect of the new terms, we can return to the situation from above, where the front of cell 1 has a junction with the back of cell 2 (see Fig 6C and 6D). Dropping all other terms, this leads to

$$\frac{d\mathbf{X}_1}{dt} = \kappa(\mathbf{X}_2^- - \mathbf{X}_1^+),$$

$$\frac{d\alpha_1}{dt} = 4\kappa\frac{r^{3/2}}{r^2+1}(\mathbf{X}_2^- - \mathbf{X}_1^+)\cdot\mathbf{e}^\perp(\alpha_1) - \mu\frac{\sqrt{r}}{r^2+1}\sin(\alpha_1 - \alpha_2).$$

Inspecting the sine-term in the equation for $\alpha_1$, we see that the cytoskeletal coupling always aids alignment, however larger aspect ratios lead to slower alignment. For the effect of the cell-cell junctions, we see that they cause the center of cell 1 to be pulled along the vector connecting the two junction points. Further, the junction also causes cell turning, however it is not

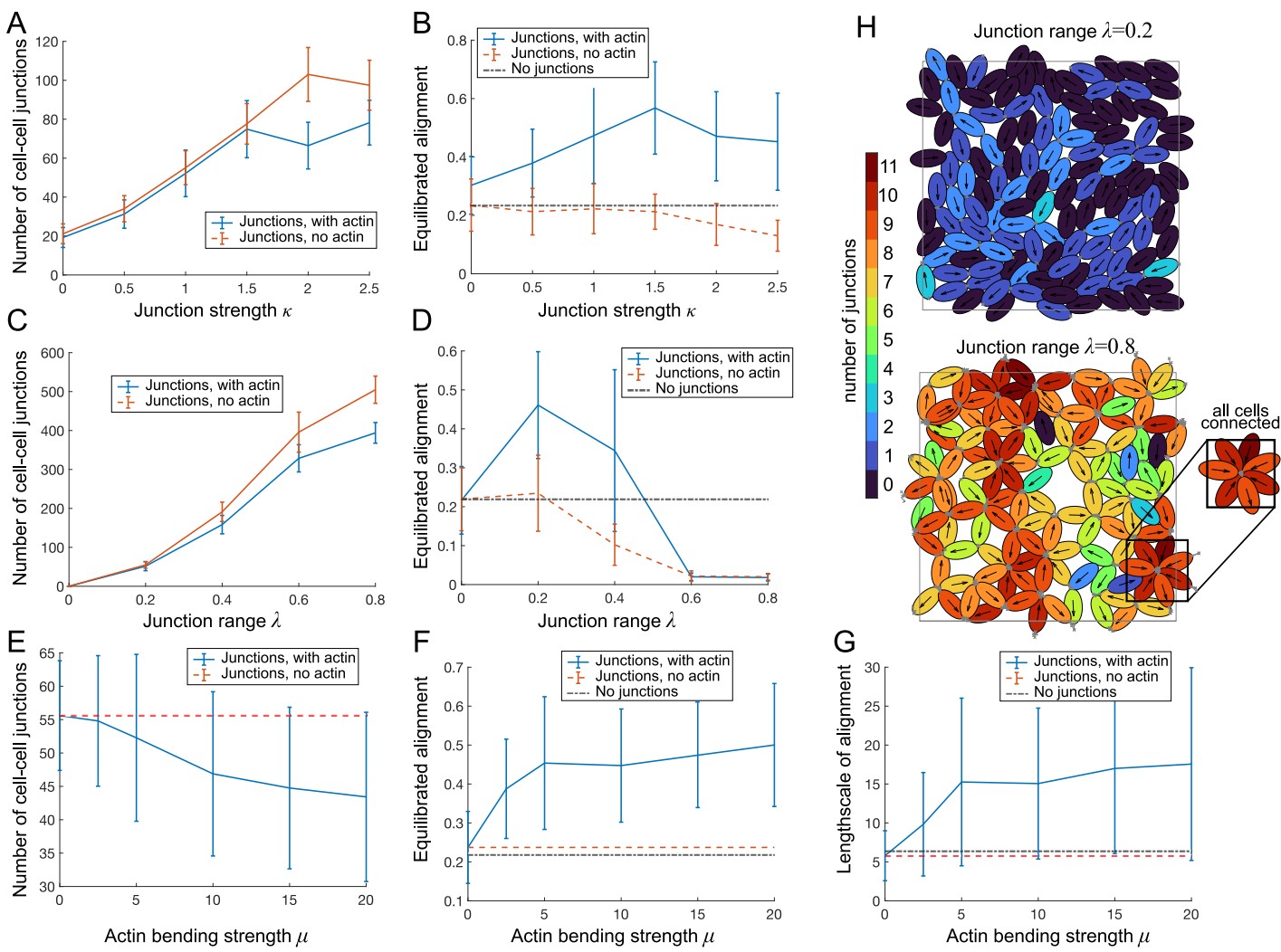

**Fig 7. A-F:** Number of cell-cell junctions (A,C,E) and value of equilibrated alignment (B,D,F) in dependence of the junction strength $\kappa$ (A,B), the junction range $\lambda$ (C, D) and the actin force $\mu$ (E,F) with and without an actin force ($\mu = 0$ and $\mu = 5$ respectively). Average over 60 simulations, error bars show standard deviation. G: Length scale of alignment in dependence on actin force $\mu$ (see S1 Appendix, Sec 2). Base parameters are $\lambda = 0.2$, $\nu = 0.5$, $\kappa = 1$, $N = 125$, $L = 20$. H: Simulation snapshots at equilibrium for $\lambda = 0.2$ (top) and $\lambda = 0.8$ (bottom). Cell junctions are marked in grey, color corresponds to number of junctions per cell. Other parameters are $\nu = 0.5$, $\mu = 0$, $\kappa = 1$, $N = 125$, $L = 20$.

obvious whether this will aid or hinder alignment. The answer becomes even less clear in a multi-cell context and in the context of overlap avoidance and self-propulsion. For this we turn to simulations.

### 4.3 Results 3: The base model with cell-cell junctions and actin forces

**Cell-cell junctions alone hinder alignment.** For cell-cell junctions forming at cell heads and tails there are two new (non-dimensional) parameters introduced to the model: The junction range $\lambda$ (i.e. maximal length over which junctions can form) and the junction strength $\kappa$. We varied $\lambda$ between 0 and 0.8 (i.e. between 0–28% of one cell length) and $\kappa$ between 0 and 2.5 (i.e. between 0–2.5 times the strength of overlap avoidance), initially without actin forces. First we measured the number of junctions formed and found, as expected, that more junctions are formed as $\lambda$ or $\kappa$ increase, Fig 7A and 7C. Next, we found that alignment decreases in both

cases, Fig 7B and 7D. It appears that the junctions hinder alignment, because they lead to cells forming clumps where more than two cells are joined at one point, which acts against alignment. Indeed, we found many more instances of cell clumps for larger junction range than for lower junction range, as demonstrated in Fig 7H.

**Supracellular actin can greatly aid alignment.** Next we investigated how adding an actin force of strength $\mu$, representing the effect of supracellular actin bundles, would affect the dynamics. We found that the actin force has only a small effect on the number of junctions formed, with a slight tendency to reduce the number of junctions formed, Fig 7A, 7C and 7E. However, we found that the actin force can greatly increase the equilibrated alignment, Fig 7B, 7D and 7F. We found that as actin forces increase, so does alignment, with alignment values plateauing for large actin forces, Fig 7F. Further we found that in the presence of actin, there is an optimal junction strength (at $\kappa \approx 1.5$, i.e the junction strength is 1.5×the strength of overlap avoidance), Fig 7B, and an optimal junction range (at $\lambda \approx 0.2$, corresponding to about 7% of one cell length), Fig 7D. Further we found that not only the value of the alignment parameter, but also its length scale, measured as defined in S1 Appendix, Sec 2, Eq (2), increases from around 5 (corresponding to 1–2 cell lengths) up to about 17 (corresponding to about 6 cell lengths). In Fig 7F and 7G we see a large variation in the alignment values and length scales measured for a given parameter set. We speculated that the reason might be differences in the populations' junction structure. This we explored next.

**High alignment at long length scales is driven by linear chains of cells.** To understand the junction structure of a cell population better, we represented the simulated cell populations as graphs, where cells and connections between them are represented by the nodes and edges of the graph respectively. This allows for a visual representation of the population structure as well quantification of graph properties, such as the number of junctions per cell (the degree of the node). For a cell to be of degree 0 means it has no connections to other cells, a degree 1 cell has a connection to one other cell, etc. For a given parameter set, we then produced 60 repetitions of the same numerical experiments (with random initial conditions) and inspected, at equilibrium, the correlation between the % of cells of a given degree with the alignment, Fig 8A, 8B and 8C. Strikingly, we found that for alignment, there is a strong negative correlation with % of degree 0 cells, a strong positive correlation with the % of degree 2 cells, and almost no correlation with the % of degree 1 cells. This means for high alignment (and high length scales of alignment) one needs few unconnected cells and many cells being connected to exactly 2 other cells. In example simulations, Fig 8D and 8E we see that, indeed, degree 2 cells form long chains that explain both the high alignment and the long alignment length scales.

## 5 Discussion

### A flexible modelling framework

In this work we have developed a framework to mechanistically model the collective behaviour of active, elliptically shaped particles, that self-propel, avoid overlap, deform and form cell-cell junctions that communicate cytoskeletal forces. The framework is based on energy minimisation and can easily be extended or adapted to include e.g. other cell shapes or different types of cell-cell interactions. A strength of the framework is that the derived equations strike a useful balance between being complex enough to capture the desired phenomena, while being simple enough to be interpretable.

We simulated the emerging collective dynamics for a large ensemble of cells and analysed the numerical results with an emphasis on alignment dynamics. The main computational findings of this work are:

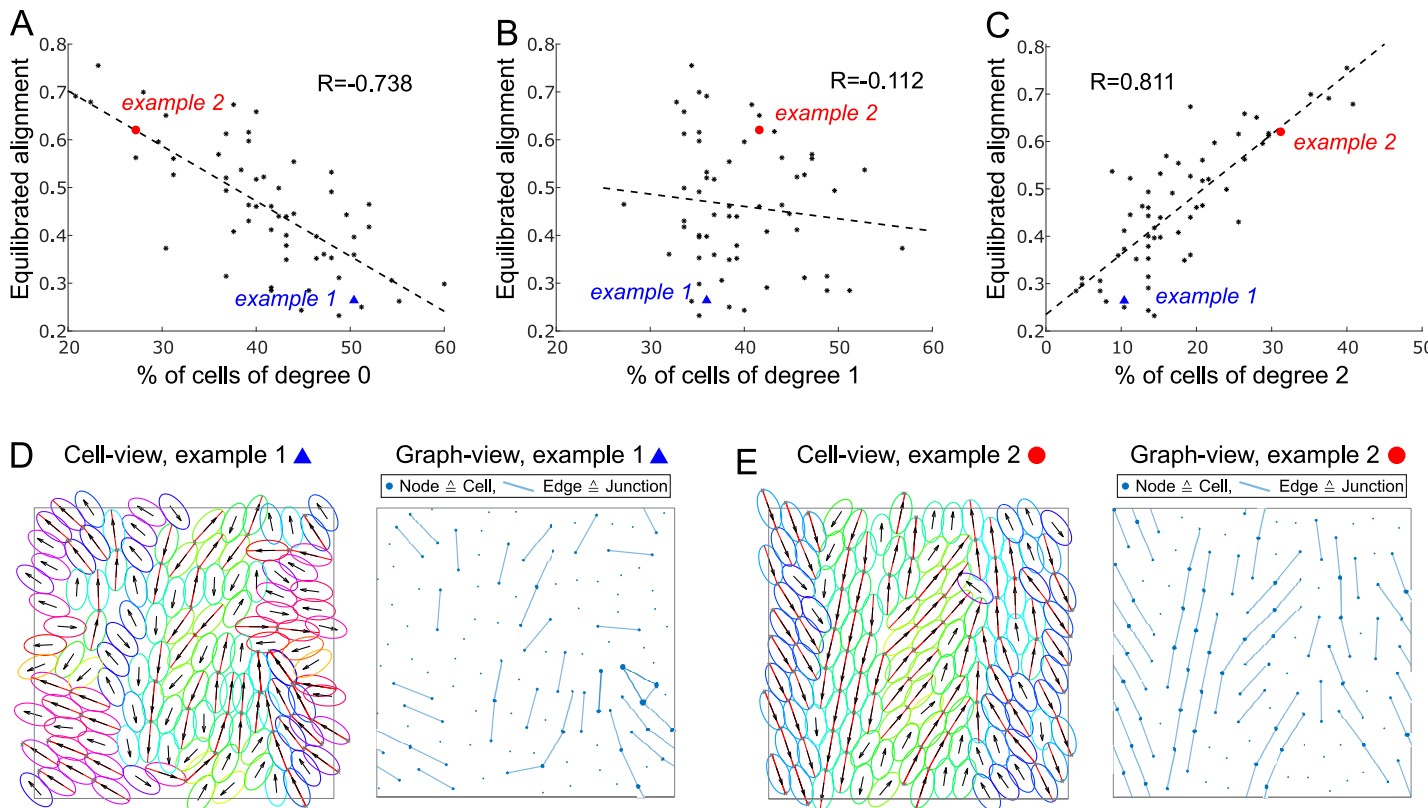

**Fig 8.** A-C: Scatter plot of equilibrated alignment against % of cells of degree 0 (A), degree 1 (B) and degree 2 (C) at the final time point $T = 400$. Each black dots represents one simulation run. The blue triangle and the red dot mark the examples in D and E respectively. R gives the correlation coefficient and the dotted line gives the linear least squares fit. D,E: Examples marked in A-C in cell-view (left, red lines mark actin, colors and arrows as in Fig 2) and graph-view (right, dots mark cells/nodes, lines mark edges/junctions). See also S3 Video. Other parameters are $\nu = 0.5$, $\mu = 5$, $\kappa = 1$, $\lambda = 0.2$, $N = 125$, $L = 20$.

- **Cell alignment needs a balance of self-propulsion and overlap avoidance.** We found that, to maximise collective cell alignment, there is an optimal ratio of self-propulsion speed and overlap avoidance. This allows cells to have a sufficiently long contacts with a sufficient number of cells, which aids alignment.

- **Deformability aids collective alignment.** We found that allowing for flexible cell shapes can aid alignment. We hypothesise that this is because it leads to to more elongated cells (which are associated with more alignment) and a more flexibility of the use of space.

- **Cell-cell junctions alone hinder alignment.** We found that modelling spring-like cell-cell junctions at the cell heads and tails hinders alignment. The reason seems to be the formation of clumps of cells.

- **Actin forces lead to strong, long-scale alignment.** If actin forces are communicated via the cell-cell junctions, this can significantly increase alignment. In this case alignment will happen on a much larger length scale. The reason seem to be long, linear chains of connected cells.

## Future work

The derived equations can be used to study e.g. the interaction of only two cells in more depth: Such a simplified system could then be analysed using analytical methods, such as stability

analysis, asymptotics or determining long-term behaviour. This is the subject of current ongoing work. The results will give further insights into the involved time scales of movement, the role of self-propulsion or the effect of deformability. Since the analysis involves a technical treatment of degenerate limits this work is reserved for a separate publication, which is in preparation. In terms of modelling, we are planning on following several directions, such as: 1) We will extend the cell-cell junction model to investigate the effect of cell-cell junctions forming along the whole cell surface. 2) We will investigate how cell-cell junctions affect cell shape. 3) We will derive and analyse a more detailed model of cytoskeletal dynamics within the cell and its interaction with the substrate. Further, we will test our insights in an experimental setting: Our work in [55], where we compared two types of fibroblasts with different overlap avoidance, is a first step in this direction. However, we will also experimentally test several of the other theoretical predictions in this work.

## 6 Material and methods

### Ethics statement

The primary dermal fibroblasts studied in this work were isolated from normal skin and keloid scar tissue from adult patients providing written informed consent. This tissue collection was ethically approved by the National Research Ethics Service (UK) (14/NS/1073). The study was conducted in accordance with the ethical standards as set out in the WMA Declaration of Helsinki and the Department of Health and Human Services Belmont Report.

For more detailed biological methods, please refer to [55]. Normal dermal fibroblasts and keloid derived fibroblasts were isolated from the collected tissue and cultured in vitro. To visualise cell overlap, cells were labelled with CellTrace reagents in two colours (Violet and CFSE, ThermoFisher). Subsequently, cells were plated on imaging substrates with a ratio of 9:1 Violet:CFSE and fixed after 24 hours. Imaging was obtained using a Zeiss LSM 880 confocal microscope (20x NA 0.8 Plan-Apochromat air objective). To evaluate cell shape, single fibroblasts were observed within confluent monolayer cultures. Mosaic expression to highlight single cells was obtained by transfecting cells with EGFP-LifeAct, followed by fixation after 48 hours, and immunostaining with phalloidin to visualise F-Actin (Life Technologies). To visualise cell-cell adhesions, cells were labelled with N-cadherin (Cell Signaling Technologies) after fixation. Imaging was obtained using a Zeiss LSM 880 confocal microscope (40x NA 1.3 Plan-Apochromat oil objective, 40x NA 1.1 LD C-Apochromat water objective, or 63x NA 1.4 Plan-Apochromat oil objective).

## Supporting information

**S1 Appendix. Derivation and computational details.**
(PDF)

**S1 Video. Video corresponding to snapshots shown in Fig 2A.** Parameters are $v = 0.2$, $N = 125$, $L = 20$, $r = 2$, $T = 800$, time step = 0.01.
(MP4)

**S2 Video. Video corresponding to snapshot shown in Fig 5D.** Parameters are $v = 0.5$, $\gamma = 0.1$, $N = 125$, $L = 20$, $\bar{r} = 2$, $T = 400$, time step = 0.01.
(MP4)

**S3 Video. Video corresponding to snapshot shown in Fig 8E.** Parameters are $v = 0.5$, $\lambda = 0.2$, $\mu = 5$, $\kappa = 1$, $N = 125$, $L = 20$, $\bar{r} = 2$, $T = 400$, time step = 0.01.
(MP4)

## Acknowledgments

The authors wish to thank Antoine Nicolas Diez for his helpful thoughts and comments and Mohit Dalwadi for his supervision support.

## Author Contributions

**Conceptualization:** Angelika Manhart.

**Formal analysis:** Vivienne Leech, Angelika Manhart.

**Funding acquisition:** Brian M. Stramer.

**Investigation:** Fiona N. Kenny, Stefania Marcotti, Tanya J. Shaw.

**Methodology:** Vivienne Leech, Angelika Manhart.

**Software:** Vivienne Leech, Angelika Manhart.

**Supervision:** Brian M. Stramer, Angelika Manhart.

**Visualization:** Vivienne Leech, Angelika Manhart.

**Writing – original draft:** Vivienne Leech, Angelika Manhart.

**Writing – review & editing:** Vivienne Leech, Stefania Marcotti, Tanya J. Shaw, Brian M. Stramer, Angelika Manhart.

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
