## [Decision Letter · Decision Letter 0]

19 Mar 2024

Dear Dr. Manhart,

Thank you very much for submitting your manuscript "Derivation and simulation of a computational model of active cell populations: How overlap avoidance, deformability, cell-cell junctions and cytoskeletal forces affect alignment" for consideration at PLOS Computational Biology.

As with all papers reviewed by the journal, your manuscript was reviewed by members of the editorial board and by three independent reviewers who are the leading experts of mathematical modeling in cell biology. While the manuscript is well-written, there are substantial concerns over the adopted methodology (e.g., the lack of treatment for randomness, cell deformability, fluidity/remodeling of cytoskeleton and cell-cell adhesion over the relevant timescales of cell alignment etc). The reviewers's comments are attached below. Since these issues are at the stem of the modeling and the conclusion, we cannot accept the paper as it currently stands.  However, we would like to invite the resubmission of a significantly-revised version with more computational results that faithfully address the reviewers' concerns.

We cannot make any decision about publication until we have seen the revised manuscript and your response to the reviewers' comments. Your revised manuscript is also likely to be sent to reviewers for further evaluation.

Sincerely,

Jian Liu

Academic Editor

PLOS Computational Biology

James O'Dwyer

Section Editor

PLOS Computational Biology

Reviewer's Responses to Questions

**Comments to the Authors:**

Reviewer #1: The authors propose an ellipsoidal cell agent-based model (ABM) for cells interacting in 2D with avoidance of overlap. The model is formulated as an energy a minimization. The aspect ratio of cells, their elastic (head-tail) tethering, and forces due to supra cellular actin cables are considered. They find that self-propulsion, along with these mechanisms aid affects the cells' ability to align with neighbours. The higher aspect ratio leads to higher alignment and packing of the cells.

The paper is well-written and interesting. I liked the step-wise approach, where the simplest cases are treated first, and then new features are introduced. I enjoyed reading it. That said, there are a few ways to improve its readability.

Major issues

1. The authors approximate cells as ellipses.

How much does this lose? Actual cells can slightly rearrange their shape to avoid having to turn or align, no? Would the conclusions hold for cells that deform by pseudopod retraction/protrusion in random directions, not maintaining exact ellipsoid shape?

2. (a) The authors suggest that (L 76-77) CPM are costly compared with other types of models. In my experience this is not the case, and there are free platforms like Morpheus that make CPM simulations very easy (and also very reproducible). As far as I could see, only the actin cables would be tricky to include. I wonder if the model conclusions would be robust to other platforms like that.

2. (b) This brings up the question of whether the model is purely deterministic? Would a little noise help or hinder alignment? [Note that CPM has an inherent energy that can be defined to avoid or promote cell-cell contact. It also has some inbuilt stochasticity. Directed cell motility can also be implemented as can a kind of preferred length and perimeter to area ratio.]

3. The paper should cite the work of Eirikur Palsson, who has a population of deformable ellipsoidal cells model with forces due to cell overlap in 3D. (He was not studying alignment, but the general idea is related.) There are also some CPM simulations with cell alignment in the literature and a free example in Morpheus that should be cited in case not already done.

4. Model presentation and equations: In general, the equations could be explained a little more clearly. Note that equations were not all numbered, so it's hard to refer to them. For example, Section 2.2 is a bit "formal" and could use a couple of sentences for biological readers. A small panel in figure 1 with your parametrization might help the non mathematician understand your energy terms better.It would also be nice to have one (or more) small panels in some figures showing the case of 2 cells where the quantities in ODEs are labeled.

While I understand that the paper is mainly directed to modellers, the current formalism presents a barrier to readers with less formal background, so a few more diagrams or text explanations could help.

5. Discussion points and some general questions:

5(a) Can the authors comment on how different is this mechanism from simple repulsion between centroids of cells? Their work probably allows some insights about the connection.

5(b) Would cell-cell junctions along the sides help alignment? I.e. can the geometry of junction placement help with alignment?

5(c) Is there a way to compare the model predictions with experimental observations? To quantify alignment and aspect ratios etc? Please correct me if I am wrong, but so far it appears that the experiments are just to show some images of cells. There is not yet any preliminary representation of the cells by ellipsoids or the alignment of those cells. How hard would that be? While this seems to be called "future work" is some preliminary comparison feasible?

5(d) The authors suggest a number of interesting future directions. One of these, namely doing the 2-cell stability analysis would seem to belong in this paper, to show some small analytical treatment. Other aspects are fine for later work.

Minor issues

Fig 1 - what are the green and magenta tracers marking?

Is X_i the cell centroid?

The small case x is not fully defined. Should it also have a subscript for cell I?

"leading to the area elements abs." - unclear

a self-propulsion speed that is independent of friction. [Is there evidence for this]?

Will cells always move in a direction of their orientation or be pushed in the direction of greatest change in overlap?

E_{overlap} - not clear what V is, and what abs refers to. Please mention that V will be discussed later.

L 149 We therefore choose V to be constant with value σ in regions of overlap and zero

elsewhere. [How hard would it be to have V depends on the contact length or overlap area?]

L 160 "We see that the cell’s center is being pushed in the direction normal to the vector connecting the points of

overlap." [Surely the model was designed this way, so it would be nice to say this earlier as a motivation for the model equations, and then confirm it here.]

L 214 For one cell we obtain the following equation for how the aspect ratio r changes over time: [explain where from, and why aspect ratio should change if this is one cell with no interactions.]

Fig 4 " Single NDF (A) and KDF".. I suggest not using abbreviations in this figure for better readability.

L 228 suggesting that shape changes

aid alignment. [Strictly speaking, this applies to changes in the aspect ratio, so unless you can assert that it is the same as arbitrary shape changes, perhaps soften the wording?]

L 243 This suggests that allowing cells to deform dynamically as a reaction to overlap introduces a more flexible way to use the available space leading to higher alignment.

[Can you support this statement with some text about what "more flexible" means?]

When cells change shape do you preserve the area (exactly, or approximately)?

L 293 junctions hinder alignment, because they lead to cells forming clumps where more than two cells are joined at one point [is this a model artifact?]

Fig 8 - I like the idea of representing the cells as a graph, but why only linear chains? Why not some branching graph?

L 329 the derived equations strike a useful balance between being complex enough to capture the desired phenomena, while being simple enough to be interpretable. [Can you comment on how much analysis can be done? Why not treat one of the simplest cases here?]

Reviewer #2: In this paper, the authors model the collective alignment of cells by incorporating mechanisms of self-propulsion and overlap avoidance. They also investigate the effects of cell deformability and cell-cell junction formation, comparing the results to experimental measurements. Overall, the paper is well-written, and the findings appear intriguing.

However, I have some significant concerns:

1) The authors present the model as a process of energy minimization per time step, but it remains unclear why this approach is chosen over defining the physics of the system through a force balance equation. Utilizing a force balance equation would provide a clearer physical principle, where frictional forces balance forces derived from a potential and active forces. This approach would likely yield the same dynamical equations, with the energy dissipated as heat.

2) In the simulations, we observe lateral sliding of cells (perpendicular to the direction of propulsion) due to pushing forces from neighboring cells. However, it's questionable whether this behavior accurately represents cell movements during alignment. Cell movement typically involves crawling along surfaces through protrusions, adhesion formation, and contractions. While the model captures some aspects of cell propulsion, it's unclear how the perpendicular sliding observed in the simulations translates to real-world scenarios.

3) The paper includes a comprehensive list of references to related works in active and motile cell systems, primarily in the introduction. However, I find a notable absence of meaningful connections between the results of this study and prior research in the broader field of polar active matter. Previous works have investigated the stability properties of homogeneous polar and nematic states, as well as mechanisms of defect formation. It's essential to contextualize the specific findings of this paper, particularly the basic model presented in Fig. 2, within the existing literature to justify the novelty and accuracy of the results.

Minor comment: The color scale in Figures such as 2A should ideally cover the range from 0 to 2pi.

Reviewer #3: The article by Leech et al. describes a well designed study on agent-based simulation of fibroblast alignment observed during wound healing. The article utilizes a propelled particle approach, and considers mechanical contacts between cells. In the 2nd half, the model incorporates the possible shape changed by the cell and potential role of actin cables. Overall, I think the paper is well organized and clearly written. However, there are no compelling results. The model formulation lacks some fundamental elements such as random forces. I would like the authors to address the following main conceptual questions.

1) The article uses a propelled particle approach. This is the most popular approach in literature, first was used by Vicsek et al. I understand that most people take it for granted that this applies to cells. In actuality, this is not too clear. In fact, it was shown pretty convincingly that cells move according to a persistent random walk model, often biased by ECM fibers (Wu et al. PNAS 2014). The next question is whether the persistent random walk model is equivalent to the propelled particle model. I think it can be shown pretty convincingly that this is not the case. I think the authors should at least mention that other models of cell movement have been proposed that are potentially fundamentally different.

2) In the context of persistent random walk model, its application to multicellular case in confluent epithelium case has been explored by Li & Sun, Biophys J. 2014 and Koride et al, APL Bioengineering 2018. These papers use the vertex model, which describes cell shape changes crudely, but nevertheless allows for some sort of mechanical interaction. I'm not sure how this compares with the elliptical approach, but should be discussed.

3) All of the models mentioned above utilizes some sort of random force. In other words, cells motion direction will randomly diffuse, ultimately generating diffusive behavior. This observation has been well established experimentally. There is no random force discussed in this work, and therefore not surprising that we see alignment. Diffusive behavior of cell motion is pretty well established, the lack of which in this work is a major question mark. I understand that authors have cell-cell interaction through actin dynamics and cell shape change, but these are not random forces. I worry that the author is missing a major ingredient.

4) The cell shape change model is reasonable but the actin cable model is not. Actin stress fiber formation is dynamic and stochastic, as has been shown extensively in biological literature. Modeling of actin stress fiber is also available. Here the authors model actin fibers as springs and rods, with even bending contribution. This is not aligned with biological observation, which shows that actin stress fibers turn over on the time scales of minutes, and dissipate and reform. Actin cytoskeleton on the time scale of cell movement is liquid like. This type of dynamics could well generate random forces that ultimately describes cell dynamics observed in this context.

5) Along the same vein, cell-cell junctions can also dynamically form and break, generating friction-like behavior between cells. Cell-cell junction should not be modeled as static linkages.

In summary, the biological relevance of the model proposed here requires more careful evaluation. The ingredients are not currently well aligned with established concepts and experiments in the field.

**Have the authors made all data and (if applicable) computational code underlying the findings in their manuscript fully available?**

The PLOS Data policy requires authors to make all data and code underlying the findings described in their manuscript fully available without restriction, with rare exception (please refer to the Data Availability Statement in the manuscript PDF file). The data and code should be provided as part

---

## [Decision Letter · Decision Letter 1]

17 Jun 2024

Dear Dr. Manhart,

Thank you very much for submitting your manuscript "Derivation and simulation of a computational model of active cell populations: How overlap avoidance, deformability, cell-cell junctions and cytoskeletal forces affect alignment" for consideration at PLOS Computational Biology. As with all papers reviewed by the journal, your manuscript was reviewed by members of the editorial board and by several independent reviewers. The reviewers appreciated the attention to an important topic. Based on the reviews, we are likely to accept this manuscript for publication, providing that you modify the manuscript according to the suggestions by the reviewer #2. 

Please prepare and submit your revised manuscript within 30 days. If you anticipate any delay, please let us know the expected resubmission date by replying to this email.  Note that for this round I will make the decision on your revised manuscript without sending it back to the reviewers. 

Sincerely,

Jian Liu

Academic Editor

PLOS Computational Biology

James O'Dwyer

Section Editor

PLOS Computational Biology

Reviewer's Responses to Questions

**Comments to the Authors:**

Reviewer #1: The authors have revised the paper well and have addressed my major concerns. The paper is ready for publication.

Reviewer #2: The authors have addressed my comments, which improved the manuscript. However, the connection between the results of this study and prior research in the broader field of polar active matter remains weak (comment 3 of my first review). Given extensive work on related systems (e.g. see review https://doi.org/10.1103/RevModPhys.85.1143 ), I recommend that the authors further improve this connection.

Reviewer #3: the authors have addressed my questions and concerns. The paper can be published.

**Have the authors made all data and (if applicable) computational code underlying the findings in their manuscript fully available?**

Reviewer #1: Yes

Reviewer #2: Yes

Reviewer #3: Yes

PLOS authors have the option to publish the peer review history of their article (what does this mean?). If published, this will include your full peer review and any attached files.

Reviewer #1: **Yes: **Leah Edelstein-Keshet

Reviewer #2: No

Reviewer #3: No

Figure Files:

Data Requirements:

Reproducibility:

References:

---

## [Editor Report · Decision Letter 2]

13 Jul 2024

Dear Dr. Manhart,

We are pleased to inform you that your manuscript 'Derivation and simulation of a computational model of active cell populations: How overlap avoidance, deformability, cell-cell junctions and cytoskeletal forces affect alignment' has been provisionally accepted for publication in PLOS Computational Biology.

Best regards,

Jian Liu

Academic Editor

PLOS Computational Biology

James O'Dwyer

Section Editor

PLOS Computational Biology

---

## [Editor Report · Acceptance letter]

22 Jul 2024

PCOMPBIOL-D-24-00196R2 

Derivation and simulation of a computational model of active cell populations: How overlap avoidance, deformability, cell-cell junctions and cytoskeletal forces affect alignment

Dear Dr Manhart,

I am pleased to inform you that your manuscript has been formally accepted for publication in PLOS Computational Biology. Your manuscript is now with our production department and you will be notified of the publication date in due course.

With kind regards,

Zsofia Freund
